# A2AR Expression and Immunosuppressive Environment Independent of *KRAS* and *GNAS* Mutations in Pseudomyxoma Peritonei

**DOI:** 10.3390/biomedicines11072049

**Published:** 2023-07-20

**Authors:** Shigeki Kusamura, Adele Busico, Elena Conca, Iolanda Capone, Luca Agnelli, Daniele Lorenzini, Silvia Brich, Marta Angelini, Chiara Costanza Volpi, Desirè Viola Trupia, Vincenzo Lagano, Tommaso Torelli, Annunziata Gloghini, Dario Baratti, Marcello Guaglio, Massimo Milione, Marcello Deraco, Federica Perrone

**Affiliations:** 1Peritoneal Surface Malignancy Unit, Fondazione IRCCS Istituto Nazionale dei Tumori, Via Venezian 1, 20133 Milan, Italy; shigeki.kusamura@istitutotumori.mi.it (S.K.); dario.baratti@istitutotumori.mi.it (D.B.); marcello.guaglio@istitutotumori.mi.it (M.G.); marcello.deraco@istitutotumori.mi.it (M.D.); 2Laboratory of Diagnostic and Molecular Research, Department of Diagnostic Innovation, Fondazione IRCCS Istituto Nazionale dei Tumori, Via Venezian 1, 20133 Milan, Italy; adele.busico@istitutotumori.mi.it (A.B.); elena.conca@istitutotumori.mi.it (E.C.); luca.agnelli@istitutotumori.mi.it (L.A.); danielelorenzini@insitutoumori.mi.it (D.L.); silvia.brich@istitutotumori.mi.it (S.B.); marta.angelini@istitutotumori.mi.it (M.A.); chiara.volpi@istitutotumori.mi.it (C.C.V.); desire.trupia@istitutotumori.mi.it (D.V.T.); tommaso.torelli@istitutotumori.mi.it (T.T.); annunziata.gloghini@istitutotumori.mi.it (A.G.); federica.perrone@istitutotumori.mi.it (F.P.); 3Medical Oncology, Fondazione IRCCS Istituto Nazionale dei Tumori, Via Venezian 1, 20133 Milan, Italy; 41st Pathology Division, Department of Diagnostic Innovation, Fondazione IRCCS Istituto Nazionale dei Tumori, Via Venezian 1, 20133 Milan, Italy; vincenzo.lagano@istitutotumori.mi.it (V.L.); massimo.milione@istitutotumori.mi.it (M.M.)

**Keywords:** pseudomyxoma peritonei, A2AR, CD39, CD73, PDL-1

## Abstract

In pseudomyxoma peritonei (PMP), *KRAS* and *GNAS* mutations are frequent. We hypothesized that these mutations may contribute to the suppression of antitumor immunity: *KRAS* may induce GMCSF expression, while *GNAS* may enhance the expression of cyclic adenosine monophosphate and A2AR signaling. This study aimed to explore possible mechanisms facilitated by *KRAS* and *GNAS* mutations for escaping immune surveillance. Additionally, we looked for new potential therapeutic and prognostic targets in this rare disease which is poorly characterized at the molecular level. GM-CSF, A2AR, CD73, CD39, and PD-L1 expression was investigated by immunohistochemistry in 40 PMPs characterized for *GNAS* and *KRAS* mutational status. Immune cell populations were studied by immunohistochemistry and nanostring nCounter^®^. Following the criteria of a prognostic nomogram reported for PMP, we stratified the patients into two different risk groups, with 28 “low-risk” and 12 “high-risk” patients. We observed the expression of GM-CSF (74%); CD39 (37%); CD73 (53%); A2AR (74%); and PD-L1 (16%) which was unrelated to *GNAS* or *KRAS* status. The tumor microenvironment showed the presence of CD4+ T cells (86%); CD8+ T cells (27%); CD20+ B (67%); CD15+ cells (86%); and CD163+ M2 macrophages (67%), while CD56+ NK cells were absent. CD163 expression (27%) in PMP tumor cells was associated with poor prognosis. *GNAS* mutation and A2AR expression were not associated with a specific immune transcriptional signature. However, the expression assay revealed 21 genes associated with prognosis. The “high-risk” patients exhibited worse progression-free survival (HR = 2.3, CI 95%: 1.1–5.1, *p* = 0.034) and significant downregulation of *MET*, *IL8*, *PPARG*, *DTX4*, *HMGA1, ZIC2*, *WNT5B,* and *CCRL2*. In conclusion, we documented the presence of immunosuppressive factors such as GM-CSF, A2AR, and PD-L1 in PMP. These factors were not associated with *GNAS* and *KRAS* status and could be explored as therapeutic molecular targets. Additionally, a set of potential prognostic biomarkers, including CD163 expression in tumor cells, deserve further investigation.

## 1. Introduction

Pseudomyxoma peritonei (PMP) is a rare clinical entity characterized by the presence of mucinous ascites, leading to significant abdominal distension and bowel obstruction [1]. PMP most commonly originates from a mucinous neoplasm in the appendix [2]. The combined treatment of cytoreductive surgery (CRS) and hyperthermic intraperitoneal chemotherapy (HIPEC) with cisplatin plus mitomycin has been established as the standard of care for PMP [3]. However, despite such aggressive intervention, recurrence rates after complete cytoreduction range from 24% to 30%. PMP relapsing or progressing after CRS/HIPEC were treated with curative-intent surgery or palliative options such as systemic chemotherapy [4].

Several studies reported a high frequency of *KRAS* (>80%) and *GNAS* (>50%) mutations in PMP, with a comutation rate >60% [5,6,7,8]. In peritoneal mucinous malignancies, *KRAS* mutation induces mucin production, synergistically engaging the *PI3K/AKT* and *MEK* pathways. Accordingly, the coinhibition of *PI3K* and *MEK* reduced mucin levels [9].

For its part, *GNAS* encodes the α-subunit of a stimulatory G-protein (Gαs), an intracellular signal transducer that activates adenylate cyclase. *GNAS* mutation induces the oncogenic activation of Gαs, resulting in increased cyclic adenosine monophosphate (cAMP) levels. Subsequently, cAMP binds to protein kinase A (PKA), leading to the release of two catalytic subunits that can phosphorylate various transcription factors, including NF-kB, HIF-1α, and CREB. The PKA/CREB signaling promotes mucin expression, and the inhibition of the cAMP/PKA pathway in PMP in in vivo models has been shown to decrease mucinous tumor growth [10].

Furthermore, both mutations may modulate immune reactions, contributing to the suppression of antitumor immunity. *KRAS* mutation, for instance, in colorectal cancer induces the expression of immune mediators such as the cytokine granulocyte macrophage-colony stimulating factor (GM-CSF), which acts as a strong immunosuppressive factor [11,12]. Additionally, cAMP derived from *GNAS* mutation plays a physiological role as an immunosuppressive messenger of T-cell function [13].

It is noteworthy that, regardless of *GNAS* mutation, Gαs is involved in “adenosine signaling,” through its coupling with the A2A-adenosine receptor (A2AR). Upon adenosine binding to A2AR, Gαs triggers the cAMP-mediated activation of PKA. Adenosine levels are regulated by the conversions of extracellular ATP into AMP and of AMP into adenosine, catalyzed by the ectoenzymes CD39 and CD73, respectively [14]. In contrast, the expression of CD39, CD73, and A2AR on tumor and stromal cells is promoted by the hypoxia- and c-AMP/PKA-mediated activation of HIF1α and NF-kB [15,16]. Owing to persistent inflammation and hypoxia in the microenvironment of solid tumors, ATP and adenosine levels may remain elevated, promoting the A2AR axis, which may further suppress antitumor immunity [17]. Specifically, A2AR signaling inhibits natural killer (NK) and CD8+ T cells and increases the generation of immunosuppressive cells, including myeloid-derived suppressor cells and tumor-associated macrophages [18,19,20].

Given the current understanding of the molecular landscape of PMP and the presence of an immunosuppressive tumor microenvironment associated with *KRAS* and *GNAS* mutations in other tumor types, this study aimed to investigate the potential role of immunosuppressive factors in PMP, allowing tumor cells to evade immune surveillance. Additionally, we aimed to explore new potential therapeutic and prognostic targets in this poorly characterized disease, which currently lacks effective treatment options.

## 2. Materials and Methods

### 2.1. Patients

This study had the approval of the International Review Board (IRB) of the Fondazione IRCCS Istituto Nazionale Tumori of Milan and was conducted on a monoinstitutional series of 40 patients with appendiceal PMP treated with complete cytoreduction (CRS) and hyperthermic intraperitoneal chemotherapy (HIPEC). This series included 31 patients from a previous study and for which the tissue material was still sufficient for new analyses [8]. The median age was 57 years (range: 27–71); the male/female distribution was 17/22; 89% had low-grade (PSOGI classification) disease [21]; and the median peritoneal cancer index was 24 (range: 3–39).

### 2.2. NGS, Immunohistochemistry, and Transcriptional Analysis

We assessed *KRAS* and *GNAS* mutational status using targeted next-generation sequencing (T-NGS) on genomic DNA extracted from formalin-fixed paraffin-embedded (FFPE) PMP samples following standard procedures, as previously described [8].

We performed immunohistochemistry (IHC) on FFPE tumor tissue sections using anti-GM-CSF (ab9741 Abcam, Cambridge, UK ); anti-adenosine Receptor A2a (ab3461 Abcam, Cambridge, UK); mAb CD73 (ab133582 Abcam, Cambridge, UK); mAb CD39 (Ab223843, Abcam, Cambridge, UK) and CD8 (C8/144B Dako, Santa Clara, CA, USA); CONFIRM anti-CD20 (L26 Ventana Medical Systems, Tucson, Arizona, AZ, USA); CONFIRM anti-CD4 (SP35 Ventana Medical Systems, Tucson, Arizona, AZ, USA); CONFIRM anti-CD163 (MRQ-26 Ventana Medical Systems, Tucson, Arizona, AZ, USA); anti-CD15 (CARB-3 Dako, Santa Clara, CA, USA); anti-CD56 (123C3 Dako, Santa Clara, CA, USA). We performed a semiquantitative analysis: we assessed the expression using a scoring system based on the immunostaining intensity (I) and the staining marker extent (E), defined as the percentage of positive cells. We combined the I and E scores into the final score I × E [22]. We classified as positive those samples with a final score ≥ 2. For PD-L1 expression, we used anti-PDL1 antibody (22C3 Dako, Santa Clara, CA, USA) and 1% as the threshold for positivity. We evaluated marker expression in tumor cells and the tumor microenvironment (TME) including stromal, immune, and endothelial cells. The Aperio ImageScope Software 12.4.3.5008 (Leica Biosystems, Wetzlar, Germany) was used for scanning slides.

We extracted RNA from FFPE tumor tissue using the miRNeasy FFPE Kit (Qiagen, Germantown, Maryland MD, USA) according to the manufacturer’s instructions. We performed transcriptional analysis by nCounter^®^ (Nanostring Technologies, Seattle, WA, USA). Specifically, we used 150 ng of mRNA and the PanCancer Pathways and the PanCancer Immune Profiling panels. The first panel quantifies the expression of 770 human genes from cancer-associated canonical pathways, including MAPK, JAK-STAT, PI3K, RAS, Cell Cycle-Apoptosis, Hedgehog, Wnt, DNA Damage-Repair, Transcriptional Misregulation, Chromatin Modification, TGF-ẞ, NOTCH, and Driver Genes. The PanCancer Immune Profiling panel quantifies the immune counterpart. We used the NanoStringNorm package for R software 4.1.2 (R Foundation for Statistical Computing, Vienna, Austria) to assess quality; the geometric mean of the counts relative to each sample, the mean plus two standard deviations and the total sum of counts options were used to correct the data for technical, background, and batch-effect issues, respectively. We used the expression counts of housekeeping genes and quantile normalization to account for intersample variations with the PanCancer Pathways Panel. We used Canberra distance and Ward as the distance metric and linkage method, respectively, to assess the natural grouping of samples. We used the *DESeq2* package for R for differential expression analyses, under standard parameters.

Gene set enrichment analysis (GSEA) and single-sample(ss)-GSEA were, finally, used for functional annotation analyses under default conditions (with weighted statistic score and 10,000 permutations of gene sets).

### 2.3. Prognosis and Statistical Analysis

Overall survival (OS) was defined as the time interval from the surgery to death from any cause, and progression-free survival (PFS) as the period from the surgery to the disease progression. We stratified patients into two different risk groups (28 “low-risk” and 12 “high-risk” patients) according to a prognostic nomogram for PMP patients based on large-scale population data [23]. Briefly, the nomogram provided an individual estimate of patient survival by incorporating demographic and disease-related parameters with critical prognostic significance.

Chi-square test was used to investigate the association between *KRAS* or *GNAS* status and GM-CSF, CD39, CD73, A2AR, PD-L1, CD4, CD8, CD20, CD15, and CD163 expression.

The Cox proportional-hazards model in the globaltest package for R was used to test the association between gene expression levels, assumed as continuous variables, and PFS or OS as clinical outcomes. Globaltest was run with 100,000 permutations on genes that varied mostly over the dataset, namely, those whose expression exceeded two variation coefficients (i.e., the standard deviation on the mean ratio). The Cox proportional-hazards model was also used to assess the correlation between PFS or OS and risk or the expression in TC or TME of the following markers: GM-CSF, CD39, CD73, A2AR, PD-L1, CD4, CD8, CD20, CD15, and CD163.

## 3. Results

### 3.1. KRAS and GNAS Status

We selected a series of 40 FFPE PMPs and successfully assessed the *KRAS* and *GNAS* mutational status using T-NGS in 37 cases. Overall, *KRAS* mutation was detected in 29 (78%), *GNAS* mutation in 19 (51%), and *KRAS/GNAS* comutation in 17 (46%) cases.

### 3.2. Immunohistochemical Analysis

The detailed data obtained from 38 PMPs (2 cases were not evaluable) are presented in Table 1, providing information on the combined scores assessed in the tumor cells (TCs) and tumor microenvironment (TME), as well as *KRAS* and *GNAS* status.

#### 3.2.1. GM-CSF

GM-CSF expression was observed in TCs in 25 out of 38 (66%) cases (Table 2), with 17 of them (68%) exhibiting strong GM-CSF expression (combined score 6, 9, or 12) (Figure 1A).

In the TME, GM-CSF expression was observed in 16 (42%) cases (Table 2), all but one showing low expression (combined score 2 or 4) (Figure 1B).

Overall, combined TC and TME analysis revealed GM-CSF expression in 28 (74%) cases.

There was no significant correlation between GM-CSF expression and *KRAS* or *GNAS* status (Table 2).

#### 3.2.2. CD39

Cytoplasmic and membranous CD39 expression in TCs was observed in 11 out of 38 (29%) cases (Table 2), with 5 of them (45%) displaying strong CD39 expression (Figure 1C).

CD39 expression in the TME was found in only three (8%) cases (Figure 1D). Notably, these three CD39+ cases were *KRAS* mutated, with two of them being *GNAS* wild type (WT) (Table 2).

Overall, either the TCs or TME were CD39 positive in 14 (37%) PMPs.

#### 3.2.3. CD73

Cytoplasmic and membranous CD73 expression in TCs was found in nine (24%) cases (Table 2), with four of them (44%) having strong CD73 expression (Figure 1E).

In the TME, CD73 expression was detected in 11 (29%) cases (Table 2), with the majority of positive cases (9/11 = 82%) displaying a low score ≤ 4 (Figure 1F).

Overall, the TCs and/or TME were CD73 positive in 20 (53%) PMPs.

#### 3.2.4. A2AR

A2AR expression was frequently observed in TCs (66%) (Table 2). Among the 25 positive cases, 20 (80%) showed strong A2AR expression (Figure 1G).

In the TME, A2AR positivity was detected in only three (8%) cases (Table 2), exhibiting a low combined score (Figure 1H).

Overall, the TCs or TME were A2AR positive in 28 (74%) PMPs and A2AR and/or CD39 and/or CD73 expression was found in 36 (95%) cases.

There was no significant correlation between CD39, CD73, or A2AR expression and *KRAS* or *GNAS* status (Table 2). However, we observed a trend of a higher A2AR expression in *GNAS* WT (14/17 = 82%) compared to mutated (11/19 = 58%) cases (*p*-value not significant). Altogether, *GNAS* mutation and/or A2AR expression were found in 33 of 38 (87%) cases.

#### 3.2.5. PD-L1

PD-L1 expression, exclusively localized to the TME, was observed in 6 of 37 (16%) cases (Figure 2A and Table 2), including 5 *KRAS*-mutated and 3 *GNAS*-mutated PMPs. One case was not evaluable for *KRAS* and *GNAS* status.

#### 3.2.6. T Cells

We successfully characterized the TME by investigating the distribution of the immune cell types in 37 cases (Table 1 and Table 3). Overall, the expression of the immune cells was not significantly associated with *KRAS* or *GNAS* status.

CD4+ T cells were observed in all but 5 (86%) cases (Figure 2B), with 14 of 32 (44%) positive cases displaying strong CD4 expression.

CD8+ T cells were detected in 10 (27%) cases (Figure 2C). Among the CD8+ T cases, all but one (90%) showed low expression (combined score 2, 3, or 4). A higher fraction of CD8+ T cells were observed in A2AR positive cases compared to negative cases, although the difference was not statistically significant (37% vs. 8%).

#### 3.2.7. Natural Killer (NK) Cells

CD56+ NK cells were absent in the TME of all cases (Figure 2D).

#### 3.2.8. B Cells

CD20+ B cells were observed in 25 (67%) cases (Figure 2E and Table 3) with the majority of cases (76%) showing low CD20 expression.

#### 3.2.9. Granulocytic Myeloid-Derived Suppressor Cells

CD15+ cells were observed in 31 (83%) cases (Figure 2F and Table 3), and 17 (55%) positive cases displayed strong CD15 expression.

#### 3.2.10. M2 Macrophages

CD163+ M2 macrophages were present in the TME of 25 (67%) cases, with 15 (60%) cases exhibiting a score of 4 or 3 (Figure 2G).

Interestingly, CD163 expression was also observed in the TCs of 10 (27%) cases, with 8 (80%) of them showing strong CD163 expression (Figure 2H). Among the 10 CD163 positive cases, all were *KRAS* mutated (34%) and 7 (37%) *GNAS* mutated, with all but one (37%) case displaying A2AR expression (37% vs. 8%; *p* = n.s.) (Table 3).

### 3.3. Transcriptional Analysis

We conducted an analysis of the gene expression profile associated with *GNAS* mutation by using the PanCancer Pathways panel. Out of 36 PMPs, 20 cases passed the quality check and were suitable for analysis. We found no statistically significant difference in expression levels between the *GNAS*-mutated and WT cases (10 vs. 10), except for the downregulated *BID* gene and the upregulated *CDKN1C, MGMT,* and *WNT10B* genes in the *GNAS*-mutated cases (Appendix A).

Next, we investigated the possible gene expression profile associated with A2AR expression in 16 A2AR positive and 6 A2AR negative PMPs. These two groups exhibited no statistically significant differences in gene expression levels, except for the downregulation of *IRS1* and *FOXL2* in the A2AR positive cases (Appendix A).

We further studied the immune cell populations in PMP using the PanCancer Immune Profiling panel. The data were acquired for 36 PMPs, and after the quality check, 33 cases were suitable for analysis. We did not observe any expression differences between the CD163 TC positive (7 cases) and negative (26 cases), *GNAS*-mutated (18 cases) and WT (15 cases), and A2AR positive (19 cases) and negative (14 cases) PMPs. GSEA analysis revealed a group of 13 patients characterized by the enrichment of genes involved in immunological and inflammation pathways. However, no significant correlation was found with *GNAS* mutation, A2AR, or CD163 expression (Appendix A).

### 3.4. Prognostic Analysis

In order to identify potential prognostic markers, we investigated the correlation between the expression of the biomarkers with overall survival (OS) or progression-free survival (PFS). The Cox proportional-hazards model showed a significant association between CD163 expression in TCs with poor PFS (HR = 2.3, 95% CI: 1–5.4, *p* = 0.047) (Figure 3A). However, in multivariate analysis, CD163 expression did not retain prognostic significance.

Although detected in a limited number of cases and not statistically significant (HR = 2.4, CI 95%: 0.92–6.3, *p* = 0.073), PD-L1 expression was predominantly observed in patients with poor PFS.

Expression of GM-CSF, CD39, CD73, A2AR, CD4, CD8, or CD20 was not associated with OS or PFS.

Regarding the transcriptional pattern, we found significant and positive correlations between the expression of FOXL2, COL4A6, and FGF14 with OS. Conversely, BLNK, BLK, USP9Y, SH2D1A, CD2, TNFRSF11A, and FOXJ1 expression showed a negative association with OS. The expression levels of LEP, TPO, PTPN5, TMEFF2, TLR9, IL1RL2, SMPD3, IL25, CCL16, and CMA1 were positively associated with PFS; whereas SH2D1A, CD2, BLNK, FCER2, and FOXJ1 were negatively associated with PFS.

### 3.5. Risk Classification

To further evaluate the prognostic significance, we classified the patients into two risk groups, namely “low-risk” and “high-risk”, using the criteria of a prognostic nomogram based on extensive population data for PMP patients [23]. In our series, we identified 28 patients in the “low-risk” group and 12 patients in the “high-risk” group, each exhibiting distinct prognostic and molecular features. In detail, in the univariate analysis, we found a statistically significant correlation between the “high-risk” group and poor PFS (HR = 2.3 C 95% CI: 1.1–5.1, *p* = 0.034) (Figure 3B). Moreover, the “high-risk” group displayed a significant downregulation of PPARG, DTX4, WINT5B, RET, MAP3K12, MET, IL8, CSF1R, HMGA1, ETS2, DUSP5, FGF9, and JUN (Appendix A). These findings highlight the molecular differences between the two risk groups and suggest their potential role as prognostic markers in PMP.

## 4. Discussion

This study aimed to investigate the presence of an immunosuppressive microenvironment potentially associated with *KRAS* and *GNAS* mutations in PMP.

Our findings revealed GM-CSF overexpression in 74% of cases with strong expression observed in TCs in 68% of positive cases. Previous research has reported the transcriptional upregulation of GM-CSF mediated by *KRAS* attributed to Ras-regulated transcription factor-binding sites in the GM-CSF promoter region [12]. Consistently, inhibition of the *PI3K* or *MAPK* pathway has been shown to abolish GM-CSF expression in *KRAS*-mutated mouse pancreatic ductal epithelial cells [12]. In our study, 69% of *KRAS*-mutated PMP showed GM-CSF positive TCs. However, it is worth noting that four *KRAS* WT cases were also GM-CSF positive, indicating that other molecular mechanisms independent of *KRAS* may contribute to GM-CSF upregulation. GM-CSF is a potent cytokine that plays a complex role in the interplay between myeloid cells and antitumor immunity or immunosuppression [24]. Specifically, the antitumor effect depends on the ability of this cytokine to promote the differentiation and maturation of antigen-presenting dendritic cells (DCs) and their migration to lymph nodes, thereby inducing T-cell activation. Conversely, high levels of GM-CSF can lead to tumor immune tolerance by impeding DC maturation or activation and promoting the generation of immunosuppressive cells such as Tregs, myeloid-derived suppressor cells (MDSCs), and granulocytes, which restrain cytotoxic T lymphocyte-mediated antitumor responses [24]. In the context of PMP, the presence of an immunosuppressive effect is suggested by the elevated levels of GM-CSF detected. This finding aligns with evidence reported in other tumors, such as mesothelioma and pancreatic ductal adenocarcinoma, where the blockade of GM-CSF with a neutralizing antibody significantly attenuated the immunosuppressive potential of myeloid cells, leading to restored T-cell proliferation and activity [12,25].

Regarding the relation between *GNAS* and adenosine signaling, our study focused on the A2AR axis, which is typically promoted by the c-AMP/PKA- and hypoxia-mediated activation of NF-kB and HIF-1α [15,17]. Our data revealed frequent and strong A2AR (80%) expression in TCs, with higher levels observed in *GNAS* WT (82%) cases compared to *GNAS*-mutated (58%) cases. Additionally, CD73 (53%) expression, predominantly with a high score, and CD39 (37%) expression were also detected. A2AR and/or CD39 and/or CD73 expression was found in 36 (95%) cases. Overall, GNAS activation either through mutation or A2AR expression was present in the majority (87%) of cases. Considering the role of adenosine signaling in suppressing antitumor immunity, A2AR expression may contribute to critical tumor-mediated immune escape in PMP. Adenosine, through the A2AR-mediated enhancement of c-AMP/PKA signaling, inhibits the activity of CD8+ T and NK cells, while stimulating MDSC and type 2 (M2) macrophages to suppress T cells and release protumorigenic cytokines [18,19,20]. Targeting A2AR or CD73 shows promise as it enhances antitumor responses in vivo through T- and NK-cell activation and it may also synergistically potentiate the effects of chemotherapy and immunotherapy in vitro [26]. Various inhibitors of A2AR are currently being investigated for cancer therapy and their clinical efficacy has been observed both as monotherapy and in combination with other agents in patients with advanced solid tumors. Importantly, the effect of A2AR inhibitors may be further enhanced by oxygenation, a novel class of antitumor drugs. These molecules can attenuate HIF-1α-mediated immunosuppression (known as oxygen immunotherapy) by reducing both extracellular adenosine accumulation and the levels of CD39/CD73 adenosine generating enzymes [27,28].

Furthermore, the PD-L1 immune checkpoint may be a potential target in PMP, as we observed PD-L1 expression in TM in a subset of cases (16%). Gleeson et al. also reported PD-L1 and PD-1 expression in 18% and 36% of PMPs, respectively [5]. This finding is significant as several clinical trials are currently evaluating the efficacy of adenosine and PD-1 coinhibition options [29].

We conducted an analysis of immune cell populations that can influence the response to immune checkpoint blockade and we found a high proportion of CD4+T (86%) and CD20+B cells (68%). The fraction of CD8+T cells (27%), mostly showing a low score, was more prevalent in A2AR positive than negative cases, although the difference was not statistically significant. Overall, these immune cell populations are strongly associated with tumor surveillance and can impact the efficacy of checkpoint inhibitors [30]. However, the absence of CD56+NK and the enrichment of CD163+M2 macrophages (67%) and CD15+ **(83%)** cells may sustain an immunosuppressive TME in PMP. Moreover, CD20+B cells, which are typically associated with antitumor immune responses, can also exhibit immunosuppressive and tumor-promoting functions [31,32,33]. Interestingly, a strong expression of CD163 was also observed in TCs in 10 (26%) cases, with a trend for a higher frequency of A2AR positive cases (37% vs. 8%).

Moreover, in a subset of PMPs, we evaluated specific transcriptional signatures using nanostring panels. Our findings suggest that neither *GNAS* nor A2AR status markedly affect gene expression in PMPs. However, we speculate that the absence of a transcriptional difference suggests that GNAS mutation and A2AR expression represent two mechanisms leading to cAMP-PKA pathway activation. Since at least one of these two alterations was present in the majority (18 of 20) of analyzed PMPs, it is not surprising that a functional annotation analysis did not reveal any specific correlations with GNAS mutation or A2AR expression. However, our analysis segregated a group of patients characterized by the enrichment of genes whose expression is enhanced in immunological (e.g., T, NK, and B functions) and inflammation (chemokines and cytokines) pathways, which may represent a potential advantage in response to checkpoint inhibitors.

We also looked for prognostic biomarkers and found that patients with CD163 positive TCs exhibited worse PFS (HR = 2.3, 95% CI: 1–5.4, *p* = 0.047), consistent with the previous literature [34]. On the other hand, nanostring experiments identified a set of 21 genes whose expression was significantly associated with OS or PFS, including COL4A6, FGF14, LEP, TPO, and PTPN5 which are involved in the PI3K, MAPK, and JAK/STAT pathways, as well as regulators of T- and B-cell stimulation/development (BLK, BLNK, SH2D1A, FCER2). These findings warrant further investigation in larger cohorts of PMP cases.

Finally, the “high-risk” group of PMPs, defined by the nomogram, exhibited poor PFS and a significant downregulation of genes mainly involved in the Transcriptional Misregulation and Hedgehog pathways. Beyond the biological implications of these genes, our data suggest that “high-risk” PMPs may have unique molecular features, supporting the potential use of transcriptional features for improving the stratification of this rare disease.

## 5. Conclusions

In conclusion, our data, although derived from a limited series of PMP, provide preliminary evidence supporting the presence of immunosuppressive factors such as GM-CSF, the A2AR axis, and PD-L1 expression in PMP. While these factors were not significantly associated with *GNAS* and *KRAS* status, they hold potential as therapeutic targets. Considering the efficacy of checkpoint inhibitor-based therapies, it is important to further investigate the potential negative predictive role played by the immunosuppressive cell fraction (CD163+M2 macrophages, CD15+cells). Concurrently, the prognostic significance of a set of potential biomarkers, especially the expression of CD163 in TCs, deserves further investigation.

## Figures and Tables

**Figure 1 biomedicines-11-02049-f001:**
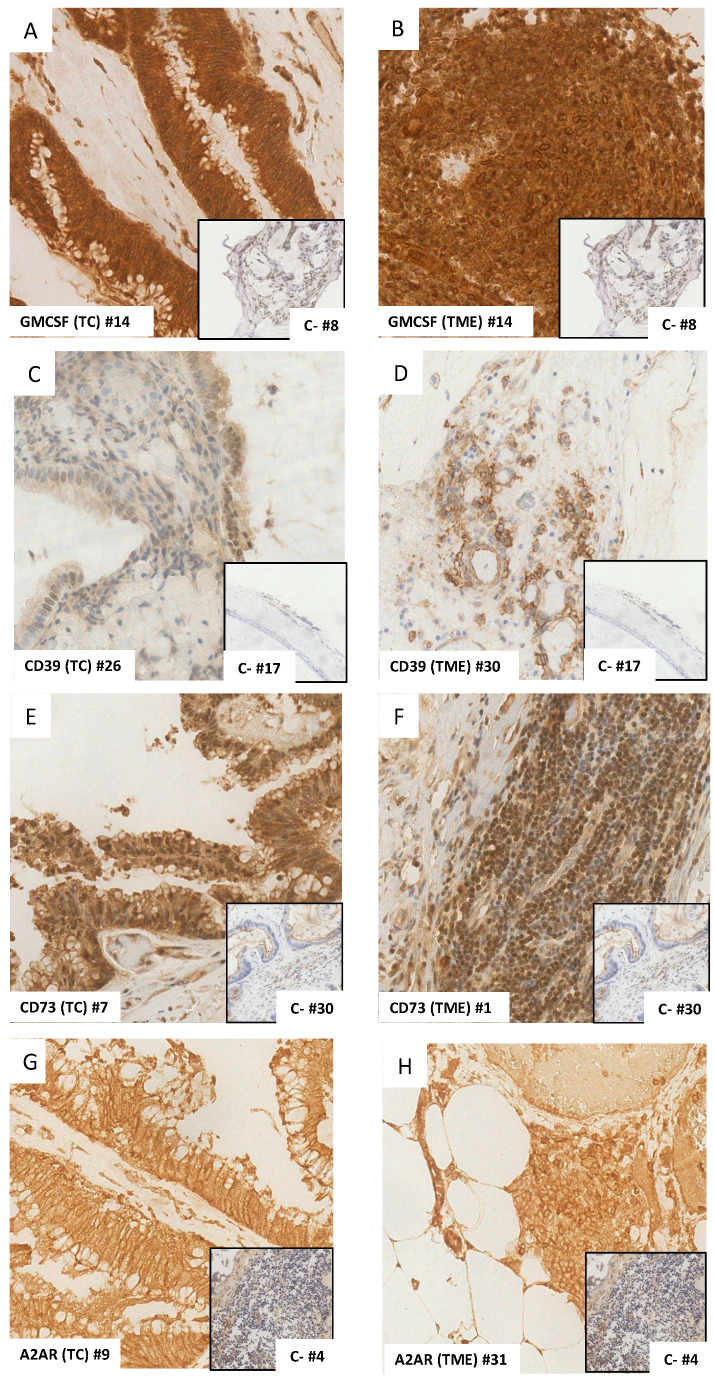
GM-CSF (**A**,**B**), CD39 (**C**,**D**), CD73 (**E**,**F**), and A2AR (**G**,**H**) expression in tumor cells (TC) and tumor microenvironment (TME) evaluated by immunohistochemistry (original magnification 40×). For each marker, the number (#) of the analyzed sample and the number of the negative control depicted in the smaller box were specified.

**Figure 2 biomedicines-11-02049-f002:**
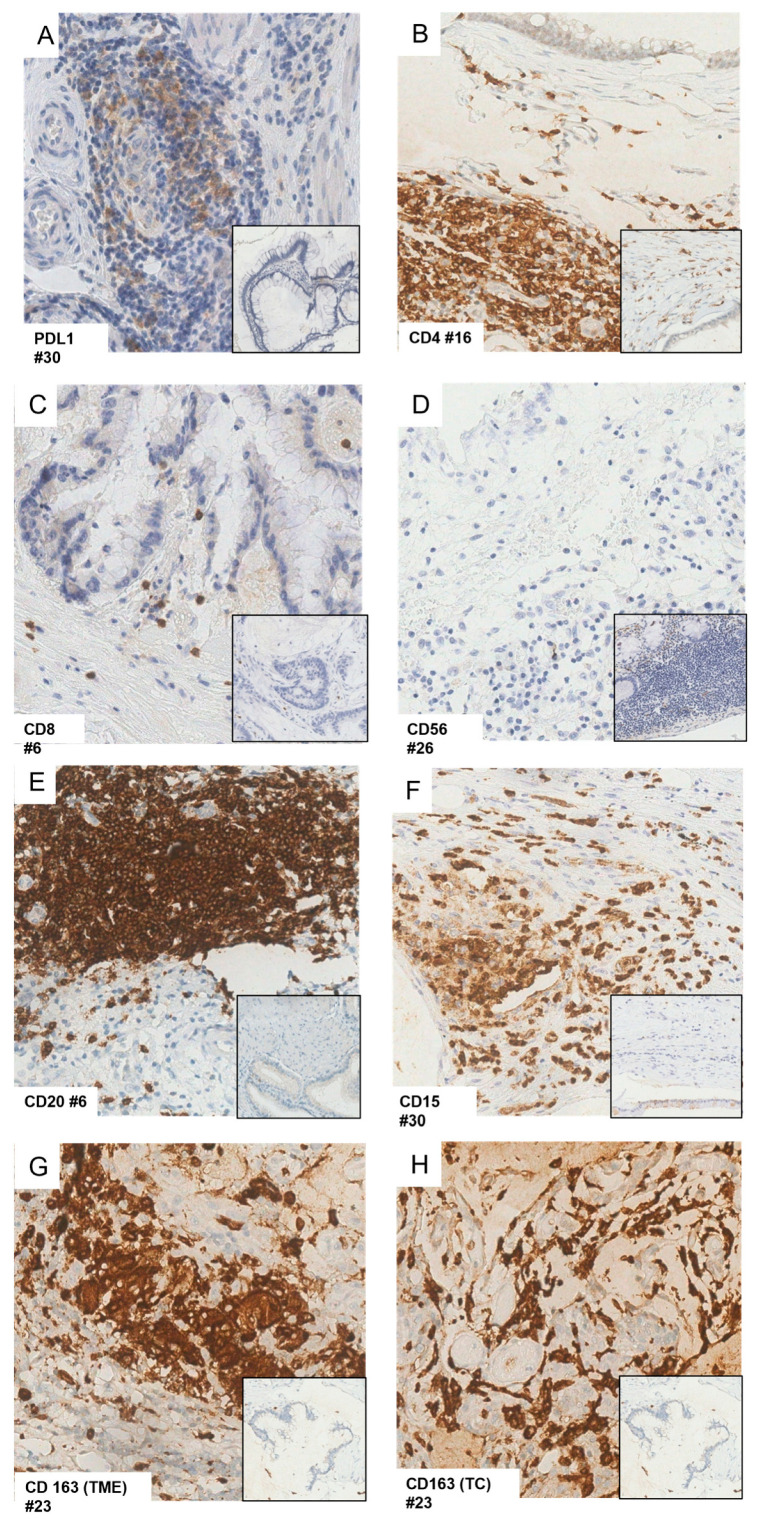
PD-L1 (**A**), CD4 (**B**), CD8 (**C**), CD56 (**D**), CD20 (**E**), CD15 (**F**) expression in the tumor microenvironment (TME); CD163 expression in the TME (**G**) and tumor cells (TC) (**H**) evaluated by immunohistochemistry (original magnification 40×). For each marker, the number (#) of the analyzed sample and the number of the negative control depicted in the smaller box were specified. For CD56, the positive control was represented by lymphoid aggregates in non-neoplastic ileum in lamina propria.

**Figure 3 biomedicines-11-02049-f003:**
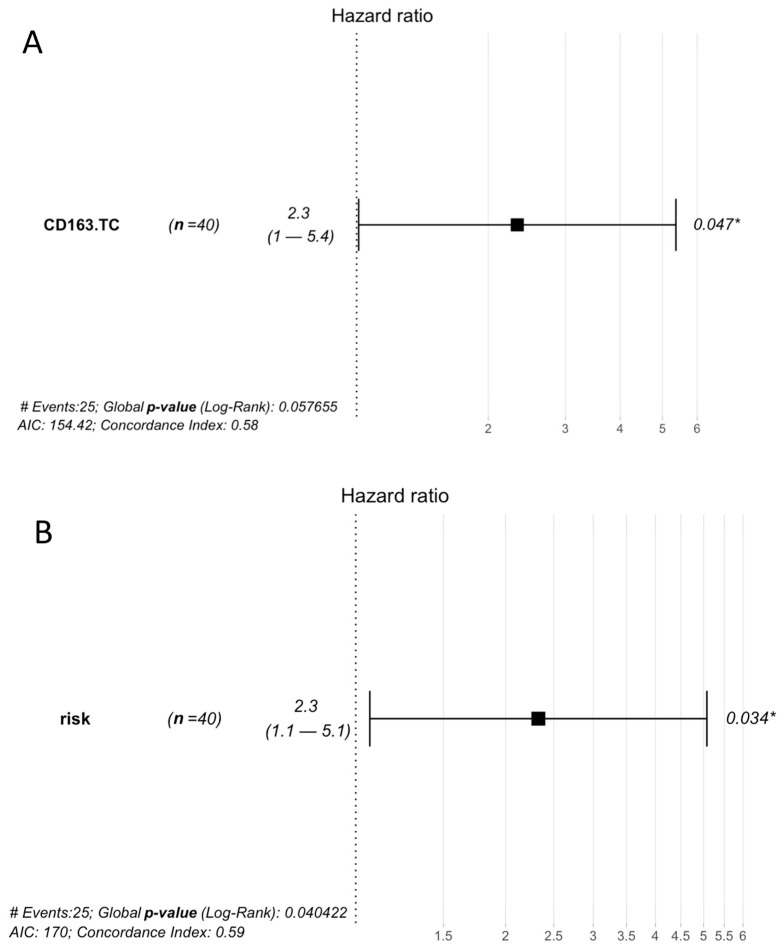
Effect size in terms of risk of death (hazard ratio) associated with: (**A**) CD163 expression in tumor cells (TC); (**B**) high-risk group classification. *: statistically significant.

**Table 1 biomedicines-11-02049-t001:** Immunohistochemistry combined scores.

No.	*KRAS*	*GNAS*	GM-CSF TC	GM-CSF TME	CD39 TC	CD39 TME	CD73 TC	CD73 TME	A2AR TC	A2AR TME	CD4 TME	CD8 TME	CD20 TME	CD56 TME	CD15 TME	CD163 TME	CD163TC	PDL1 TME
1	M	WT	3	2	0	0	0	4	6	0	9	0	2	0	0	2	9	0
2	M	M	12	0	4	0	3	0	4	0	4	0	2	0	6	0	0	0
3	WT	WT	3	0	0	0	0	0	4	0	6	0	2	0	6	6	0	0
4	WT	WT	0	0	0	0	0	4	0	0	4	0	2	0	3	2	0	0
5	M	WT	0	2	0	0	0	0	6	0	9	0	9	0	0	2	6	0
6	M	WT	9	0	0	0	0	0	6	0	4	4	6	0	3	4	0	0
7	WT	WT	9	2	3	0	6	0	9	0	NE	NE	NE	NE	NE	NE	NE	NE
8	M	WT	0	0	0	0	0	0	6	0	4	0	2	0	6	4	0	0
9	WT	M	9	2	0	0	6	0	9	0	4	2	0	0	0	0	0	0
10	NE	NE	2	0	0	0	4	0	0	0	2	0	2	0	2	6	0	0
11	M	M	4	0	0	0	4	0	0	0	4	0	4	0	0	2	0	0
12	M	M	6	4	0	0	0	0	9	0	6	2	9	0	4	6	6	0
13	M	M	12	2	0	0	0	4	12	0	6	0	2	0	3	0	0	0
14	M	WT	9	4	0	0	9	0	2	0	4	0	4	0	2	4	0	0
15	M	M	0	4	0	0	0	0	4	0	3	2	12	0	9	2	3	0
16	M	M	9	4	4	0	0	0	12	0	6	3	6	0	12	3	12	0
17	M	M	0	0	0	0	0	4	0	0	4	0	0	0	2	4	0	0
18	M	M	6	0	0	2	4	0	0	0	0	0	0	0	3	0	0	0
19	WT	WT	6	4	4	0	0	0	9	0	9	0	6	0	6	4	0	0
20	M	WT	9	2	9	0	0	4	9	0	0	4	0	0	6	0	0	POS
21	NE	NE	0	4	0	0	0	2	0	3	4	0	0	0	6	4	0	POS
22	M	WT	4	0	2	0	0	0	4	0	4	2	4	0	0	0	0	0
23	M	M	0	0	0	0	0	0	0	0	6	6	4	0	2	4	9	0
24	M	M	9	2	4	0	0	4	6	0	6	2	4	0	4	0	12	POS
25	M	M	4	9	0	0	0	0	6	0	4	0	0	0	6	2	0	POS
26	M	M	9	0	6	0	0	0	9	0	6	0	4	0	0	4	12	0
27	M	M	6	2	0	0	0	2	9	0	12	3	4	0	4	4	12	POS
28	M	WT	12	2	9	0	0	0	9	0	4	0	0	0	6	0	0	0
29	M	M	0	0	0	0	0	3	0	0	4	0	2	0	9	3	0	0
30	M	WT	0	0	0	9	0	0	0	4	9	0	4	0	6	0	4	POS
31	M	M	0	0	0	0	0	0	0	3	3	0	0	0	6	4	0	0
32	M	M	4	0	9	0	0	0	9	0	0	0	0	0	6	4	0	0
33	M	WT	4	0	9	0	0	0	6	0	3	0	3	0	6	2	0	0
34	WT	WT	0	0	0	0	0	0	0	0	0	0	0	0	3	0	0	0
35	M	WT	12	0	0	4	0	4	9	0	9	0	2	0	6	4	0	0
36	M	WT	9	0	0	0	9	0	6	0	0	0	0	0	6	0	0	0
37	WT	M	0	0	0	0	0	6	0	0	6	0	4	0	2	4	0	0
38	M	M	0	0	0	0	2	0	0	0	2	0	0	0	6	0	0	0

Abbreviations: M: mutation; WT: wild type; NE: not evaluable; POS: positive.

**Table 2 biomedicines-11-02049-t002:** Distribution of GMS-CSF, CD39, CD73, A2AR, and PD-L1 immunohistochemical expression according to *KRAS* and *GNAS* mutational status. Markers were evaluated both in the tumor cells (TC) and tumor microenvironment (TME).

		38 CASES	29 *KRAS* MUT	7 *KRAS* WT	19 *GNAS* MUT	17 *GNAS* WT
GM-CSF+	TC	25 (66%)	20 (69%)	4 (57%)	12 (63%)	12 (70%)
	TME	16 (42%)	12 (41%)	3 (43%)	8 (42%)	7 (41%)
CD39+	TC	11 (29%)	9 (31%)	2 (28%)	5 (26%)	6 (35%)
	TME	3 (8%)	3 (10%)	0	1 (5%)	2 (12%)
CD73+	TC	9 (24%)	6 (31%)	2 (28%)	5 (26%)	3 (18%)
	TME	11 (29%)	8 (27%)	2 (28%)	6 (31%)	4 (23%)
A2AR+	TC	25 (66%)	21 (72%)	4 (57%)	11 (58%)	14 (82%)
	TME	3 (8%)	3 (10%)	0	1 (5%)	2 (12%)
PD-L1+	TC	0	0	0	0	0
	TME	6 (16%)	5 (17%)	0	3 (16%)	2 (12%)

TC: tumor cells; TME: tumor microenvironment; M: mutation; WT: wild type; POS: positive: ne: not evaluable; legend: TC: tumor cells; TME: tumor microenvironment; MUT: mutation; WT: wild type.

**Table 3 biomedicines-11-02049-t003:** Distribution of CD4, CD8, CD56, CD20, CD15, and CD163 immunohistochemical expression according to KRAS and GNAS mutational status and A2AR expression. Markers were evaluated in the tumor microenvironment (TME) only, except for CD163, which was evaluated both in tumor cells (TC) and tumor microenvironment (TME).

		37 CASES	29 KRAS MUT	6 KRAS WT	19 GNAS MUT	16 GNAS WT	24 A2AR POS	13 A2AR Neg
CD4+	TME	32 (86%)	25 (86%)	5 (83%)	17 (89%)	13 (81%)	21 (87%)	11 (85%)
CD8+	TME	10 (27%)	9 (31%)	1 (17%)	7 (37%)	3 (19%)	9 (37%)	1 (8%)
CD56	TME	0	0	0	0	0	0	0
CD20+	TME	25 (67%)	20 (69%)	4 (67%)	12 (63%)	12 (75%)	18 (75%)	7 (54%)
CD15+	TME	31 (84%)	24 (83%)	5 (83%)	16 (84%)	13 (81%)	19 (79%)	12 (92%)
CD163+	TME	25 (67%)	19 (65%)	4 (67%)	13 (68%)	9 (56%)	16 (67%)	9 (69%)
	TC	10 (27%)	10 (34%)	0	7 (37%)	3 (18%)	9 (37%)	1 (8%)

TC: tumor cells; TME: tumor microenvironment; POS: positive; Neg: negative; MUT: mutation; WT: wild type.

## Data Availability

Code and data are fully available upon request.

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
