# Peer review of "A2AR Expression and Immunosuppressive Environment Independent of KRAS and GNAS Mutations in Pseudomyxoma Peritonei"

_biomedicines, 2023, doi:10.3390/biomedicines11072049_

Round 1

Reviewer 1 Report

In my opinion, the reviewed work is interesting and valuable. One weakness is the relatively small patient group, but this can be justified by the rarity of the disease. However, before making a decision on publishing the work, the following issues should be taken into account:

1. The abstract of the paper begins with a reference to the search for new therapeutic methods. Therefore, the introduction should include information about the mechanisms of action and effectiveness of currently used drugs.

2. Please indicate the aim of the study precisely. I suggest including a problem statement and a summary in relation to the aim of the study.

3. Please consider including information about the analyzed mutations in the title of the paper (as they are a significant point of consideration in this publication).

4. Please clearly indicate the results of the correlation analysis between mutations and the factors under investigation.

5. In the discussion, please describe the potential applications of the obtained results in the search for new pharmacotherapy methods.

6. The manuscript requires language and editorial corrections.

Moderate editing of the paper is required.

Author Response

  1. The abstract of the paper begins with a reference to the search for new therapeutic methods. Therefore, the introduction should include information about the mechanisms of action and effectiveness of currently used drugs.   We added information about treatments and their efficacy.

2. Please indicate the aim of the study precisely. I suggest including a problem statement and a summary in relation to the aim of the study. As suggested we better specified the aim of the study both in the abstract and introduction.

  1. Please considerincluding information about the analyzed mutations in the title of the paper (asthey are a significantpoint of consideration in thispublication). As suggested we modified the title of the paper : A2AR expression and immunosuppressive environment independent of KRAS and GNAS mutations in pseudomyxoma peritonei
  2. Please clearly indicate the results of the correlation analysis between mutations and the factors under investigation. As suggested, in the abstract and discussion we indicated this correlation. In the Results, this correlation was already present, however we made it clearer.
  3. In the discussion, please describe the potential applications of the obtained results in the search for new pharmacotherapy methods. We think that in the discussion this aspect is described. Indeed, we described the therapeutic potential of GM-CSF neutralizing antibody, as well as the availability of A2AR inhibitors alone or in combination with other drugs including PD-1 co-inhibition.
  4. The manuscript requires language and editorial corrections. We edited english language.

Reviewer 2 Report

This manuscript deals with the analysis of a cohort of pseudomyxoma peritonei (PMP) aimed to characterize the tumor microenvironment.  The analysis performed are immunohistochemistry (IHC) and nanostring nCounter.

The topic is of interest as this tumor is quite rare and the analysis of a large cohort of patients can give relevant insight on the topic.

The aim of the manuscript is to define the composition of the microenvironment, and it appears that some markers typical of immune-regulating cells are expressed.

The IHC does not allow the precise definition of cells which express a given immune-regulating antigen. In some instances, a double staining or stripping and re-probing should be applied to give more information on the cell populations which express a given antigen. More relevant could be polychromatic immunofluorescence and digital analysis (or even better, imaging mass cytometry).  

The analysis of the images could be performed with appropriate software for image analysis, but this not indicated in the methods. 

The absence of CD56+ cells should be also supported, showing that the antibody is indeed working on CD56+ cells in IHC performed by the authors on positive tissue specimens where NK cells are indeed present.

Paragraph 3.2.7. It is reported in several papers that CD56+ NK cells are cytotoxic when express this antigen at low level together with CD16. Furthermore, it is reported that usually NK cells in the TME are CD56 bright and CD16 negative and less cytotoxic. The sentence "NK cells have great cytolytic activity leading to potent anti-tumor activity" should be changed to be in line with what reported in the literature.

Moreover, many other markers expressed by NK cells can be analysed to better identify NK cells.

The IHC analysis for different markers, it is clearly performed on different tumor specimens. The figure legends and the figures themselves do not allow identifying on what anonymized samples the IHC has been performed.

It should be fine to have an idea of which markers are present in a given specimen, and also a quantitative analysis could be performed (always using appropriate software).

This should be clearly stated in each image and figure legend.

 In the present form, the manuscript appears preliminary and the absence of any functional experiment determine that the message is not really strong.

I understand well the difficulty of the enrolment of samples, but in this form the manuscript indicate that some potential regulatory cells are present in the tumor microenvironment. I would say that this analysis could be really much more in depth compared to what the authors have made.

The images in general are of poor quality (they appear quite blurry), and they should be substituted with better ones.

Negative controls for each staining should be included in the panel. In some instances, the staining appear unspecific (for example, the staining with CD56 (fig2 D) seems quite specific, although the authors stated that no CD56 are present). Furthermore, the staining with GMCSF, CD73 and A2AR is quite diffuse, and it appears that anything is stained. I would consider quite difficult to understand positive and negative samples with these kinds of staining.

The supplementary files could be shown in the manuscript.

The results section is a description of the expression of each marker analyzed that renders the manuscript not fascinating and readable.

I think that English language is enough good to understand the message of the paper. I am not so qualified to define if the manuscript is well written. However, the description of the IHC is quite tedious and not so interesting.

Author Response

This manuscript deals with the analysis of a cohort of pseudomyxoma peritonei (PMP) aimed to characterize the tumormicroenvironment.  The analysis performed are immunohistochemistry (IHC) and nanostringnCounter.

The topicis of interest as this tumor isquite rare and the analysis of a large cohort of patients can give relevant insight on the topic.

The aim of the manuscript is to define the composition of the microenvironment, and it appears that some marker stypical of immune-regulating cells are expressed.

1.The IHC does not allow the precise definition of cells which express a given immune-regulatingantigen. In some instances, a double staining or stripping and re-probing should be applied to give more information on the cell populations which express a given antigen. More relevant could be polychromaticimmunofluorescence and digital analysis (or evenbetter, imaging mass cytometry).  We agree with the reviewer that more in-depth results would be obtained with the methods he suggests. However, in this study we carried out a preliminary surveyto identify the main aspects to be explored in a second research.

2.The analysis of the images could be performed with appropriate software for image analysis, but this not indicated in the methods. We specified in the methods the software Aperio Image Scope used for image analysis. 

3.The absence of CD56+ cells should be also supported, showing that the antibody is indeed working on CD56+ cells in IHC performed by the authors on positive tissue specimens where NK cells are indeed present. As suggested, we indicated a CD56 positive control.

4.Paragraph 3.2.7. Itis reported in several papers that CD56+ NK cells are cytotoxic when express this antigen at low level together with CD16. Furthermore, it is reported that usually NK cells in the TME are CD56 bright and CD16 negative and less cytotoxic. The sentence "NK cells have great cytolytic activityl eading to potent anti-tumoractivity" should be changed to be in line with what reported in the literature. Wethank the reviewer for this observation and we removed the sentence. We did not comment the cytotoxic activity of NK because the immunohistochemistry is notable to identify CD56 bright /dim and CD16 negative NK cells.

5.Moreover, many other markers expressed by NK cells can be analysed to better identify NK cells. This an interesting point that should be investigated in an other series of PMP.

6.The IHC analysis for different markers, it is clearly performed on different tumor specimens. The figure legends and the figures themselves do not allow identifying on what anonymized samples the IHC has been performed. We specified the analyzed samples  in the figures and the table 1.

7.It should be fine to have an idea of which markers are present in a given specimen, and also a quantitative analysis could be performed (always using appropriate software). As suggest weadded the table 1 that describes all amples and allmarkers, with the combined scores assessed by an expert pathologist who classified the samples by using a  semi-quantitative analysis.

8.This should be clearly stated in each image and figure legend. We specified the sample in the figure.

  1. In the present form, the manuscript appears preliminary and the absence of any functional experiment determine that the message is notr eally strong. We agree with the reviewer that our results wereobtained in a limited, even if precious, series of PMP, so are preliminary results.  We added a sentence at the end of the discussion. However, due to few tumor cells in abudant mucus of PMP, the material that can be analyzed is limited and it makes functional experiments verydifficult.

10.I understand well the difficulty of the enrolment of samples, but in this form the manuscript indicate that some potential regulatory cells are present in the tumormicroenvironment. I would say that this analysis could be really much more in depthc ompared to what the authors have made.  The peculiar features of this rare tumor (poor tumor cells and abudant mucus) make difficult a depth analysis. However, in a new series of PMP we will be able to design further experiments on the basis of these preliminary data.

11.The images in general are of poor  quality (they appear quite blurry), and they should be substituted with better ones . As suggested we improved the quality of the images by using Aperio Image scope software. Among the author we added Dr Daniele Lorenzini, the pathologist who makes new figures.

12.Negative controls for each staining should be included in the panel. In some instances, the staining appear unspecific (for example, the staining with CD56 (fig2 D) seems quite specific, although the authors stated that no CD56 are present). Furthermore, the staining with GMCSF, CD73 and A2AR is quite diffuse, and it appears that any thing is stained. I would consider quite difficult to understand positive and negative samples with thes ekinds of staining. In each figures we added a negative control; for CD56 we added a positive control.

13.The supplementary files could be shown in the manuscript.  We added the 2 supplementary figures in the manuscript.

14.The results sectionis a description of the expression of each marker analyzed that renders the manuscript not fascinating and readable. We are aware that the description of the markers is not verya ppealingin the results. As far a spossible, we have lightened the description by removing the list of IHC scores which have instead been detailed in the new table 1.  We realized that the CD15 data were not correctly tabulated so they were reviewed by the pathologist and we updated Table 3 and the results. 

Comments on the Quality of English Language

  1. I think that English language is enough good to understand the message of the paper. I am not so qualified to define if the manuscript is well written. However, the description of the IHC is quite tedious and not so interesting. We edited english language

Reviewer 3 Report

In this study, authors have searched new therapeutic targets in pseudomyxoma peritonei (PMP), exploring possible escape mechanisms to immune surveillance mediated by KRAS and GNAS mutations. The best findings allow to conclude that in PMP, immunosuppressive actors such as GM-CSF, A2AR axis and PD-L1 are present and possibly exploitable as therapeutic targets.

The work appears well done and the results seem quite convincing to support the conclusions.

About the formal aspect, the numbering of the references both in the text and in the list seems very strange and never seen before. Is it correct or an editorial oversight?

Author Response

In this study, authors have searched new therapeutic targets in pseudomyxomaperitonei (PMP), exploring possible escape mechanisms to immune surveillance mediated by KRAS and GNAS mutations. The best findings allow to conclude that in PMP, immunosuppressive actors such as GM-CSF, A2AR axis and PD-L1 are present and possibly exploitable as therapeutic targets.

The work appears well done and the results seem quite convincing to support the conclusions.

About the formal aspect, the numbering of the references both in the text and in the list seems very strange and never seen before. Is it correct or an editorial oversight?   We apologize for the editorial mistake. In the revised version, the references have been formatted with the correctly updated EndNote output style.

Round 2

Reviewer 2 Report

Although the authors did not perform additional experiments requested, they made some corrections that can be considered enough to endorse the manuscript for publication.

English is good.